# Hydrogel Dressing Biomaterial Enriched with Vitamin C: Synthesis and Characterization

**DOI:** 10.3390/ijms251910565

**Published:** 2024-09-30

**Authors:** Piotr Szatkowski, Zuzanna Flis, Anna Ptak, Edyta Molik

**Affiliations:** 1Department of Biomaterials and Composites, Faculty of Materials Science and Ceramics, AGH University of Krakow, Al. Mickiewicza 30, 30-059 Krakow, Poland; pszatko@agh.edu.pl; 2Department of Animal Biotechnology, Faculty of Animal Science, University of Agriculture in Krakow, Al. Mickiewicza 24/28, 31-059 Krakow, Poland; flis.zuzanna@gmail.com; 3Laboratory of Physiology and Toxicology of Reproduction, Institute of Zoology and Biomedical Research, Faculty of Biology, Jagiellonian University, Gronostajowa 9, 30-387 Krakow, Poland; anna.ptak@uj.edu.pl

**Keywords:** biodegradable polymers, hydrogel dressings, bioactive substances, vitamin C

## Abstract

Materials engineering has become an important tool in the field of hydrogel dressings used to treat difficult-to-heal wounds. Hydrogels filled with bioactive substances used as a targeted healing system are worthy of attention. Vitamin C has healing and supporting effects in the treatment of many skin problems. The aim of the research was to produce a hydrogel biomaterial enriched with ascorbic acid for use as a dressing for difficult-to-heal wounds. A total of four different dressings were developed, each with different modifications in each layer. The dressing with vitamin C in the third layer was shown to release vitamin C ions more slowly than the dressing with vitamin C in the first layer. The studies conducted have shown that the dressings containing vitamin C have, among other things, a higher compressive strength, are characterised by a lower relative shortening after the application of force and shorten without damage at a lower force than in the case of a dressing without vitamin C. The dressings designed have a very good stability in the temperature range of 18 °C to 60 °C. It was found that the higher the vitamin C content in the dressing, the greater the increase in the specific heat value of the transformations. Therefore, hydrogel dressings containing vitamin C may be candidates for local delivery of vitamin C to the skin and protection of the wound area.

## 1. Introduction

Difficult-to-heal and chronic wounds are a problem that has been with people since time immemorial [1]. Wound management is associated with chronic pain and discomfort for patients [1]. Currently, the problem of chronic wounds such as diabetic wounds or pressure ulcers is increasing due to the prevalence of lifestyle diseases and an ageing population. To accelerate the wound healing process, herbal preparations or natural substances containing mainly animal fats, milk, honey or egg white were initially used [1,2]. In the 20th century, George D. Winter made a breakthrough in the history of wound dressings. He showed that the use of an active dressing in a moist environment doubled the rate of wound healing [3]. This discovery has led to the development of many generations of wound dressings, and the latest third-generation dressings are based on hydrogels and hydrophilic polyurethane films [3]. Currently, poor wound healing can be caused by many factors, such as post-operative complications. Impaired wound healing is common in patients with diabetes, in the elderly with leg ulcers and in patients with pressure ulcers [3]. In addition, non-healing or slow-healing wounds have a significant impact on patients’ quality of life, causing pain and discomfort and often leading to social isolation, depression and financial stress. In recent years it has become important to develop functional dressings that can protect the wound area, promote healing and protect the wound from infection. Hydrogel dressings are used to treat difficult-to-heal wounds such as burns, post-operative wounds and pressure ulcers [1,3]. According to T.D. Turner, an ideal dressing should have the following functions: provide an optimal environment for wound healing, absorb wound exudate, not adhere to the wound, cleanse the wound of dead tissue, be antibacterial and hypoallergenic, contain no toxic ingredients and allow oxygen diffusion to accelerate cellular activity [3].

Materials engineering has become an important tool in the field of dressings as it deals with the dysfunction of the damaged tissue or organ. It creates biocompatible materials that are used in tissue regeneration with the aim of restoring normal biological functions. With the development of biomaterials and the advancement of tissue engineering technology, some innovative tissue engineering dressings and scaffolds, such as nanofibres, films, foams and hydrogels, have been widely used in the field of biomedicine, especially in wound treatment [4]. Among them, hydrogels have attracted much attention due to their unique advantages [4]. The introduction of hydrogel dressings in regenerative medicine accelerates wound healing. Hydrogels are characterised by good ductility, high water content and favourable oxygen transport, making them one of the most promising materials for wound dressings [5]. The water contained in hydrogels has a transport function. Nanomaterials show better biodegradability, biocompatibility and colloidal stability in wound healing and may play a role in promoting healing due to their properties or as carriers of other bioactive substances [5]. In addition, hydrogels filled with bioactive agents have the potential to treat wounds as a targeted therapeutic system [6]. By adding drugs to hydrogels, we can create an intelligent drug delivery system where the rate of drug release can be freely manipulated [7]. The incorporation of drugs into hydrogels is possible because hydrogels swell in response to various factors such as environmental pH, temperature, ionic strength or the presence of antibodies [8]. Hydrogel dressings loaded with antibiotics can deliver the drug directly to the infected site. Hydrogel dressings are widely used in the treatment of wounds that are difficult to heal, including burns, pressure ulcers and post-operative wounds. The advantage of hydrogel dressings is their transparency, which allows observation of the wound without the need to remove the dressing [9]. Because of their flexibility, biocompatibility and ease of design, hydrogels are widely used in the manufacture of dressings [10]. In recent years, the use of biomaterials has attracted increasing attention as a valuable material for the design of biomedical dressings that can perform a specific bioactive function and then be degraded without polluting the environment [6,11]. Hydrogel dressings are therefore a more suitable alternative to typical plasters and bandages, which are most commonly used in the treatment of skin wounds due to the need for frequent replacement with a new dressing and the lack of biodegradability. Hydrogel dressings inhibit water evaporation and also provide a breeding ground for bacteria, which are retained in the dressing structures as they migrate from the wound into the gel. They also absorb excess wound exudate, providing a barrier against infection. Hydrogel dressings relieve pain and do not stick to the wound, so they do not damage tissue during dressing changes [9,10].

Ascorbic acid has healing and supportive effects in the treatment of many skin problems, including pressure ulcers, foot ulcers, corneal ulcers, burns and mechanical injuries [12,13]. Vitamin C is a key factor in the wound healing process and therefore delayed and impaired healing of subcutaneous tissue is associated with its deficiency [14]. The major extracellular protein in the granulation tissue of a healing wound is collagen. Ascorbic acid is a water-soluble vitamin responsible for the synthesis of connective tissue, particularly collagen [12,14]. This process involves hydroxylation of selected amino acids (proline and lysine) in the newly synthesised procollagen protein and specific enzymes called hydroxylases are responsible for these important reactions. Hydroxylase enzymes require ascorbic acid and iron as cofactors [12]. Vitamin C also provides tensile strength to newly formed collagen [14]. In addition to its role in collagen synthesis, vitamin C also increases fibroblast proliferation and angiogenesis [15]. It is also crucial in the formation of cross-links between collagen fibres, which promotes collagen synthesis [16]. Vitamin C also enhances ceramide synthesis, creating strong barrier lipids in the epidermis. In addition, ascorbic acid has immune functions, protects the skin from ultraviolet radiation and damage, helps in the synthesis of neurotransmitters and also acts as a major antioxidant in the body [12,15]. Ascorbic acid has a healing effect, providing the right environment and conditions for faster skin regeneration [17]. Wound infections are associated with an alkaline pH due to the presence of bacteria, while creating an acidic environment around the wound promotes faster healing. In addition, inflammation caused by skin injury is thought to accelerate the breakdown and consumption of vitamin C in tissues [13]. It is therefore important to provide vitamin C at the wound site to initiate epidermal repair processes. Local administration of vitamin C reduces inflammatory responses and accelerates wound healing, particularly in burn wounds [18]. Studies by Lima et al. (2009) on the treatment of skin wounds in rats with ascorbic acid cream showed that ascorbic acid acts at every stage of the wound healing process [17]. It reduced the number of macrophages, increased the proliferation of fibroblasts and new vessels, and stimulated the synthesis of thicker and more organised collagen fibres in the wounds [17]. Hydrogel dressings are currently a highly sought-after product in the field of biomedicine. The introduction of hydrogel dressings with natural ingredients allows the innovative use of bioactive substances in dressing materials. Hydrogel dressings with added bioactive substances can be used for different types of wounds and individual patient needs, increasing their effectiveness. Hydrogel dressings enriched with ascorbic acid, which, in addition to its effective local action in the healing process of skin wounds, is a cheap material and can reduce healthcare costs, can increase the effectiveness of therapy and improve patient comfort [12,19].

Hydrogels are commonly used as a material for cell encapsulation. Hydrogels can be prepared from natural biomaterials such as alginate, chitosan, hyaluronic acid, or synthetic materials such as polyvinyl alcohol, polyacrylamide and polyethylene glycol [20,21]. One of the most commonly used polymers is sodium alginate. Alginates are bioresorbable polysaccharides of natural origin. Alginates are used in biomaterial engineering because of their high biocompatibility. Cells or tissues are immobilised in a hydrogel structure, which protects them from mechanical damage. The gel structure reduces stress on the cellular elements, which are very delicate. Another advantage of hydrogel capsules is their ease of manufacture. To protect the hydrogel structure, the capsules are additionally surrounded by semi-permeable membranes. Encapsulation of cells in hydrogel capsules has been used in the treatment of diabetes—the islets of Langerhans have been encapsulated [22]. Fiorentini et al., 2023, developed a plant-based microfibrous scaffold filled with vitamin C. Studies have shown that the released vitamin C was able to stimulate the production of collagen mRNA over time [11]. In addition, plant-based hydrogel microfibres with vitamin C reduced the expression of inflammatory cytokines in the wound environment [11]. Combining biopolymers with substances with wound-healing properties may provide opportunities for the synthesis of matrices that stimulate and trigger target cell responses that are critical to the healing process [23]. In recent years, many researchers have used a variety of techniques to produce wound dressings, resulting in products with different properties [23]. Currently used wound dressings include hydrogels, films, nanofibres, foams, topical preparations, transdermal patches, sponges and bandages [24]. Many biopolymers such as collagen, cellulose, chitosan and alginate are also used to make dressings [24,25,26]. Each method and material has its own strengths and limitations, so the search for the best hydrogel dressing for hard-to-heal wounds should continue [23].

Therefore, for many of the reasons outlined above, the aim of the research carried out was to produce a hydrogel biomaterial enriched with ascorbic acid and to characterise it by thermogravimetric studies, DMA and DSC analysis, elasticity, static compression and membrane tests.

## 2. Results

### 2.1. Thermogravimetric Measurements

Thermogravimetric measurements were performed for pure polyvinyl alcohol (PVA); L-ascorbic acid (vitamin C); hydrogel dressing without added vitamin C (dressing no. 1); dressing containing gauze, hydrogel and nanotubes; dressing containing polylactide (PLA) film with two layers of gauze; hydrogel dressing with added vitamin C in the first (top) hydrogel layer (dressing no. 2); hydrogel dressing with added vitamin C in the second (bottom) hydrogel layer (dressing no. 3).

Figure 1 shows the results of the thermogravimetric tests. The TGA analysis of the sample containing PVA shows that the polymer starts to degrade at 253 °C and reaches its maximum degradation at 272 °C. The total percentage weight loss is 65.393%. The moisture content in the raw PVA was approximately 1%. These results indicate that despite the high hygroscopicity of PVA, the moisture content can be reduced to a level that allows the production of repeatable hydrogel matrices from this polymer.

Further studies illustrating the course of thermogravimetric analysis for pure vitamin C (ascorbic acid) have shown that degradation begins at about 193 °C. The mass derivative curve shows two main peaks, the first associated with the degradation of vitamin C and the second, which reaches its maximum at 315 °C, associated with the complete degradation of vitamin C. At this temperature, the vitamin C sample loses over 35% of its mass (Figure 2). The data obtained confirm that vitamin C are characterised by high thermal stability (up to 200 °C) and low hygroscopicity.

Thermogravimetric studies of dressing layers containing PVA, nanotubes and gauze have been carried out and have shown that the mass change curve has three peaks. The first mass change occurs up to 147 °C and is related to the evaporation of water contained in the dressing (breaking of hydrogen bonds in the hydrogel network formed by water molecules). The second change, from 147 °C to 215 °C, is caused by the degradation of the cotton fibres that make up the gauze. The third peak, which reaches its maximum at 296 °C, is related to the degradation of the PLA polymer matrix (Figure 3).

Studies of the structural substrate consisting of cotton gauze and polylactide film showed a main peak during which the tested sample loses approximately 85% of its mass (Figure 4). The low mass content of the cotton embedded in the PLA matrix causes the temperature of the onset of cotton degradation to be spread over a wider temperature range. The stability of this structural layer of the designed dressing is 115 °C.

Studies carried out on three different dressings showed that for dressing no. 1, which did not contain vitamin C in its layers, the mass derivative curve for this dressing has three main peaks. The first peak results from the loss of water from the sample, the second is related to the degradation of the PVA matrix and the third is related to the degradation of the PLA film with cotton gauze (Figure 5).

In the subsequent dressings containing vitamin C, they were more durable and the peaks were shifted to the left compared to the dressing without vitamin C (Figure 6 and Figure 7).

Thermogravimetric studies have shown that temperature is a significant factor in material degradation (Table 1). It has been shown how individual elements in the dressing affect the degradation of the dressing as a whole and at what temperature they lose 1–5% of their mass.

### 2.2. Test—Differential Scanning Calorimetry (DSC)

Studies of the results of differential scanning calorimetry for a dressing without vitamin C in its layers have shown that the first stage, observed as a stepwise increase in specific heat in the temperature range from −39 °C to 30 °C, is associated with the melting of the water contained in the dressing. This increase is approximately 82.3 J/g and reaches its maximum at a temperature of 2.4 °C. The second peak occurs in the temperature range from 30 °C to 171 °C and is associated with the evaporation of water contained and bound in the dressing; in the first stage, strongly bound water evaporates, then weakly bound water evaporates in the structure of the dressing—this is evidenced by two jumps in this stage. The increase in specific heat in the range of these temperatures is about 891 J/g. The last stage, in the temperature range of 172 °C to 187 °C, is related to the degradation of the polylactic acid present in the sample. The increase in specific heat during PVA degradation is about 5.3 J/g. (Figure 8).

The result of differential scanning calorimetry for a dressing containing vitamin C in two layers of hydrogel is shown in Figure 9. Similar to Figure 8, three increases in specific heat were obtained. The first stage occurs in the temperature range of −20 °C to 33 °C and is associated with the melting of water. The increase in specific heat is approximately 109.8 J/g and reaches its maximum at a temperature of 7.7 °C. The second peak, which occurs in the temperature range from approximately 40 °C to 167 °C, is associated with the evaporation of water contained in the dressing. The increase in specific heat in the range of these temperatures is approximately 986 J/g. The final stage, in the temperature range of 171 °C to 181 °C, is associated with the degradation of the PVA polymer contained in the dressing. The increase in specific heat during PVA degradation is approximately 5.4 J/g.

Figure 10 shows the results of the DSC analysis for a dressing containing vitamin C in a layer of hydrogel. Similar to the dressings in the previous graphs, three increases in specific heat were obtained. The first stage, which occurs in the temperature range from −24 °C to 39 °C, is related to the melting of water. The increase in specific heat is approximately 84.1 J/g and reaches its maximum at a temperature of 15.7 °C. The second increase in specific heat occurs in the temperature range from about 40 °C to 176 °C and is related to the evaporation of water contained in the dressing. The increase in specific heat in the range of these temperatures is about 772 J/g. The last stage, in the temperature range of 175 °C to 187 °C, is related to the degradation of the layer containing the PLA polymer. The increase in specific heat during PLA degradation is about 0.99 J/g.

Table 2 shows the values of the start temperature of the endothermic process (T_0_), the end temperature of the endothermic process (T_k_), the maximum temperature (T_max_) and the specific heat (S) in the given temperature ranges.

### 2.3. Dynamic Mechanical Analysis (DMA) Studies

DMA analysis of a dressing sample containing no vitamin C in its layers showed that the loss modulus (E″) increases rapidly at temperatures from −40 °C to 18 °C (which is related to the presence of frozen water); after exceeding 18 °C, it increases slightly and remains at a similar level up to 61 °C, and at temperatures above 61 °C, it begins to decrease rapidly. The modulus of elasticity (E′) remains at a similar level throughout the temperature range, being most stable between 19 °C and 60 °C (Figure 11).

On the other hand, the DMA analysis for the dressing containing vitamin C in both hydrogel layers showed that the loss modulus (E″) increases rapidly at temperatures from −40 °C to 15 °C (due to the presence of water crystals); after exceeding 14 °C, it increases linearly, remains at a similar level until reaching 71 °C, and at temperatures above 71 °C it begins to decrease rapidly. The modulus of elasticity (E′) remains at a similar level throughout the temperature range, being most stable between 15 °C and 72 °C (Figure 12).

Analysing the data obtained, it was found that the dressings gave the best results in the temperature range of 18 °C to 60 °C as the modulus of elasticity and the loss modulus remained at similar levels and were stable (Table 3).

Based on the data in Table 4, the percentage increase in the value of the modulus of elasticity (E″) as a function of the vitamin C content of the dressing was calculated using the formula:$$U=\frac{\left|{E\prime \prime }_{with\;vit\;C}-{E\prime \prime }_{without\;vit\;C}\right|}{{E\prime \prime }_{without\;vit\;C}}\cdot 100\%$$

Substituting:$$U_{18{}^{\circ}C}=\frac{3107150\;Pa-464199\;Pa}{464199\;Pa}=569.3\%$$
$$U_{60{}^{\circ}C}=\frac{3266330\;Pa-562676\;Pa}{562676\;Pa}=480.5\%$$

Comparing the results for loss modulus and elastic modulus for two types of dressing, without vitamin C and with vitamin C in both layers, it was shown that the addition of vitamin C improves the elastic properties of the material by one unit. In the temperature range from 18 °C to 60 °C, the bandage with vitamin C is more stable, as shown by the angle of inclination of the graphs to the straight line parallel to the horizontal axis (Figure 13. DMA studies show that it is very unfavourable for dressings to be exposed to an environment below 0 °C. This causes the water loosely bound to the hydrogel matrix and the water already present in the hydrogel structure to freeze, creating a three-dimensional network. Despite the freezing of the hydrogel, the hydrogel takes on a gel consistency once it exceeds 0 °C.

**Table 4 ijms-25-10565-t004:** Results for the compression test of selected dressings.

No.	Sample Name		Relative Shortening [mm] for a Given Force					
		Force [N]	10	20	30	40	50	100
1.	Dressing-ctrl		0.26 ± 0.05	0.45 ± 0.08	0.57 ± 0.13	0.65 ± 0.03	0.75 ± 0.04	0.99 ± 0.09
2.	Dressing no. 2		0.22 ± 0.04	0.49 ± 0.06	0.75 ± 0.09	1.01 ± 0.10	1.10 ± 0.11	1.42 ± 0.13
3.	Dressing no. 3		0.29 ± 0.05	0.52 ± 0.06	0.70 ± 0.08	0.88 ± 0.08	1.02 ± 0.09	1.29 ± 0.10
5.	Dressing no. 5		0.34 ± 0.06	0.55 ± 0.07	0.68 ± 0.09	0.77 ± 0.08	0.84 ± 0.09	1.08 ± 0.09

### 2.4. Mechanical Strength Test—Static Compression Test

Four samples of dressings containing vitamin C in different layers were subjected to compression tests. The results of the compression tests for each type of dressing are shown in the graph (Figure 14) and in Table 4. By analysing the graph, we can determine how the presence of vitamin C affects the strength of the dressings.

A force value of 10 N corresponds to hand pressure. The smallest relative shortening for this force occurs for dressing no. 2, which contains vitamin C in both hydrogel layers. All dressings containing vitamin C shorten without damage in the force range of approximately 20 N. Damage to the dressing containing no vitamin C in its layers occurs at a force of more than 40 N.

### 2.5. Membrane Test

The membrane study was carried out on three different dressings. Dressing-crtl (control), which does not contain vitamin C in its layers, dressing no. 3, which contains vitamin C in the first layer, and dressing no. 4, which contains vitamin C in the third layer. Table 5 shows the data obtained during the study.

Based on the data from Table 5, two graphs (Figure 15 and Figure 16) have been produced showing the changes in pH and conductance over time for the different dressings.

## 3. Discussion

Hydrogels can effectively support cell proliferation and facilitate wound healing due to their hydrophilic nature and three-dimensional network structure. In addition, they are characterised by good biodegradability, biocompatibility, adhesion, air permeability and the ability to maintain a moist environment for cell migration [27]. Studies have been carried out on layered dressings because the ability to use layers provides control over the rate of penetration of active ingredients between layers. Multifunctional hydrogel dressings can not only provide physical protection and maintain moisture in the microenvironment but can also improve the healing process by influencing the stage of wound healing [28]. PVA-based hydrogel dressings are well suited as a matrix for dressings incorporating functional molecules into their structure (providing a specific activity depending on the specificity of the wound being treated). Vitamin C, a potent antioxidant, mimicked typical organic molecules in the studies that contain different functional groups in their structure (which allow, enhance or hinder migration in the hydrogel matrix). Antioxidant hydrogels can remove excess reactive oxygen species in wounds, reducing oxidative stress and thus improving the wound microenvironment, promoting collagen synthesis and lowering wound pH to accelerate healing and reduce infection [29]. Studies using the membrane have provided information on the behaviour of dressings under conditions similar to those found in the skin. It was shown that dressing no. 4, which contained vitamin C in the third layer, released vitamin C ions more slowly than dressing no. 3, which contained vitamin C in the first layer, as evidenced by a slower increase in conductance and a slower decrease in pH.

Polyvinyl alcohol (PVA) is a common polymer characterised by good biocompatibility, non-toxicity and excellent mechanical properties [30]. However, pure PVA hydrogels do not have haemostatic or antibacterial effects and do not exhibit elasticity [31]. Among the various hydrogels described in the literature, hydrogels prepared from PVA mixed with some natural and synthetic additives have been shown to be very effective and are the most common route for membrane synthesis due to their high biocompatibility and ease of chemical derivatisation or modification [32]. Therefore, in recent years, scientists have focused on combining PVA with other functional components to accelerate wound healing [31]. Studies by Kamoun et al. (2015) have shown that hydrogels based on PVA composite polymers have superior properties to their counterparts made from other polymers [32]. The compression strength tests performed showed that the highest value of the parameter was characteristic of the dressing containing vitamin C in two layers. The dressings containing vitamin C in their layers were characterised by a lower relative shortening after the applied force, but they shortened without damage at a lower force than the dressing without vitamin C.

An extremely important factor in characterising a hydrogel dressing for use on different types of wounds is its thermal stability, which depends on the polymer material used and any additives. For practical reasons, it is very important to design hydrogel dressings that actively function at temperatures above approximately 25 °C. Roy et al. (2013) prepared hydrogels for biomedical applications in combination with synthetic and natural polymers such as polyvinylpyrrolidone (PVP), carboxymethylcellulose (CMC) and boric acid (BA) to obtain antimicrobial properties [33]. These hydrogels were stored at 5 °C ± 2 °C, 22 °C ± 3 °C and 40 °C for 180 days. Studies have shown that the temperature of 5 °C ± 2 °C is the best for PVP-CMC and PVP-CMC-BA hydrogels [33]. Such a low dressing stability temperature can be problematic due to the human body temperature (36.6 °C) and the temperature of the injured skin environment, which is higher due to wound inflammation. For this reason, such a dressing may not be active in the wound environment. Liu et al. (2021) designed a hydrogel dressing by combining polyacrylamide, gelatin and ε-polylysine [34]. In the temperature range of −20 °C to 60 °C, all the desired properties of the hydrogel, including superstretchability, permanent water retention, adhesion and permanent antibacterial properties, were stable [34]. In addition, the resulting hydrogel showed good heat resistance. The dynamic mechanical analysis we carried out showed that the dressings had very good stability in the temperature range of 18 °C to 60 °C (no change in mechanical properties despite the force applied with frequency). It is at these temperatures that the dressings performed best. The addition of vitamin C to the dressing improves the storage modulus value. As the temperature falls below 18 °C and rises above 60 °C, the ability of the material to absorb energy decreases. The content of vitamin C in the dressing improves the viscoelastic properties, and the dressing containing vitamin C is more stable.

The thermogravimetric (TG) study provided information on the change in sample mass for the dressing components and for the dressings produced. The main loss in mass is caused by the loss of water from the dressing. The addition of vitamin C to dressings increases the residue at the end of the study. Hydrogel dressings should have sufficient adhesion and appropriate mechanical properties to adhere to and completely seal wounds under moist and dynamic conditions [35]. The ability to retain moisture within the dressing is also extremely important, providing moisture at the wound site and allowing cell migration and proliferation [36]. The DSC study shows the effect of vitamin C addition on dressing properties. The higher the vitamin C content in the dressing, the greater the increase in specific heat in the transformations. The DSC study, together with the DMA study, shows a large effect of the water content in dressings. In fact, the high melting energy and high evaporation energy of water in the DSC study are also reflected in the storage modulus curve in the DMA study.

In summary, studies have shown that the properties of dressings vary depending on the number of layers of vitamin C and also the location of the layers of vitamin C in the dressing. In general, the proposed layered dressings are characterised by high stability, durability and effective release of ions into the wound environment. The studies presented are of an applied nature. However, further research should be carried out to determine the most optimal design of a layered hydrogel dressing with ascorbic acid in terms of usability and efficacy. Currently, hydrogel dressings for wounds are available on the market, but the authors have no knowledge of the available hydrogel dressings with ascorbic acid used for difficult to heal wounds. The designed layered hydrogel dressings enriched with vitamin C seem to be an innovative solution and could be a breakthrough in the market because of the low cost and ease of production and because of the effectiveness in supporting the treatment of difficult to heal wounds.

## 4. Materials and Methods

### 4.1. Preparation of the Hydrogel System

#### 4.1.1. Hydrogel Matrix

To prepare a 4% solution of polyvinyl alcohol (PVA), 8 g of pure polyvinyl alcohol (Sigma Aldrich, St. Louis, MO, USA) was mixed with 192 cm^3^ of distilled water. The solution was stirred on a magnetic stirrer in a water bath at 60–80 °C until the alcohol dissolved, approximately 30 min. To prepare a 4% borax solution, 1.6 g of anhydrous sodium tetraborate borax (Sigma Aldrich) and 38.4 g of distilled water were mixed. The solution was stirred with a rod at approximately 400 °C until a clear solution was obtained.

#### 4.1.2. Functional Layer

Carbon nanotubes (NanoA-mor, Los Alamos, NM, USA) were added to a 4% solution of polyvinyl alcohol. Then, 1/3 of the solution obtained was poured into a special trough and gauze was placed on the poured solution. The procedure was repeated twice. The prepared dressing layer had two layers of sterile gauze (sterilised cotton layer). To make a scaffold for the dressings, 200 mL of dichloromethane (DCM) and 25 g of poly-lactide (PLA) granules were mixed together. The solution was then mixed on a magnetic stirrer without heating, with the lid slightly tightened, until a clear solution was obtained. The solution obtained was then poured onto the trough and covered with a layer of gauze. The prepared layer was left to evaporate the DCM. In the next step, to prepare the hydrogel, the previously prepared 4%/8% solution of polyvinyl alcohol (PVA) was mixed with a 4% solution of borax. The process of PVA gelation using the above solutions consisted of linking the PVA chains via the hydrogen and coordination bonds formed [37] (Figure 17).

Vitamin C (ascorbic acid) was chosen to test the migration of a biologically active molecule characterised by the presence of many functional groups (including hydroxyl groups) in the hydrogel matrix. Carbon nanotubes were used to simulate a hydrophobic modifier of the hydrogel matrix, which weakens the affinity of the hydroxyl group in ascorbic acid and accelerates the migration process of active substances in the tested layered dressing.

### 4.2. Combining Layers into a Hydrogel Dressing

In the second stage of the study, a dressing consisting of five layers was produced: the first and third layers were hydrogels modified with vitamin C, the second and fourth layers were hydrogels with gauze and nanotubes, the fifth layer was a construction substrate containing a layer of gauze and PLA films (Figure 18):−First layer—hydrogel based on 8% PVA solution + vitamin C modification.−Second layer—hydrogel based on 4% PVA solution with nanotubes and gauze.−Third layer—hydrogel based on 8% PVA solution + vitamin C modification.−Fourth layer—hydrogel based on 4% PVA solution with nanotubes and gauze.−Fifth layer—polylactide film with a layer of gauze.

From the prepared layers, four types of dressings were made with different vitamin C contents in the layers:Dressing No. 1—no vitamin C content in the first and third layers (control dressing)

/1st layer—pure hydrogel/2nd layer—hydrogel with CNT and gauze/3rd layer—pure hydrogel/4th layer—hydrogel with CNT and gauze/5th layer—hydrogel with CNT and gauze/PLA film with gauze;

Dressing No. 2—vitamin C content in the first and third layers

/1st layer—hydrogel with vitamin C/2nd layer—hydrogel with CNT and gauze/3rd layer—hydrogel with vitamin C/4th layer—hydrogel with CNT and gauze/5th layer—hydrogel with CNT and gauze/PLA film with gauze;

Dressing No. 3—vitamin C content in the first layer

/1st layer—hydrogel with vitamin C/2nd layer—hydrogel with CNT and gauze/3rd layer—pure hydrogel/4th layer—hydrogel with CNT and gauze/5th layer—hydrogel with CNT and gauze/PLA film with gauze;

Dressing No. 4—vitamin C content in the third layer

/1st layer—pure hydrogel/2nd layer—hydrogel with CNT and gauze/3rd layer—hydrogel with vitamin C/4th layer—hydrogel with CNT and gauze/5th layer—hydrogel with CNT and gauze/PLA film with gauze.

### 4.3. Test Methods for Manufactured Dressings

#### 4.3.1. Thermogravimetry (TG)

The DTG curve determines the change in the rate of decomposition of a substance with an increase or decrease in temperature. TG analysis can be used to determine the effect of individual components on the thermal stability of the dressing [38]. Thermogravimetric analysis was performed using a TGA 550 (TA) instrument. The tests were carried out at room temperature with a heating rate of 10 °C/min in the temperature range of 25–400 °C. The paper presents the method of measurement and the results of thermogravimetric tests for seven samples: three samples are different types of hydrogel dressings, and the remaining samples are single components of dressings.

#### 4.3.2. Dynamic Mechanical Analysis (DMA)

Thermal analysis of dynamic mechanical properties (DMA) allows the determination of the temperature dependence of the dynamic modulus of elasticity E* and its components: the storage modulus E′ and the loss modulus E″. Changes in the tangent of the mechanical loss angle as a function of temperature characterise changes in the molecular mobility of the tested system as a function of temperature [38]. During the DMA test, the sample is subjected to sinusoidal deformations. The DMA analysis was performed using the DMA 850 (TA) instrument for two types of dressings: one containing vitamin C in both hydrogel layers and one without the addition of vitamin C. The temperature range during the test was from −40 °C to 120 °C, and the temperature was increased at a rate of 5 °C/min. The amplitude of the vibrations was 50 um, the frequency was 10 Hz and the compressive force was 2 N.

#### 4.3.3. Differential Scanning Calorimetry (DSC)

The study was carried out using a Mettler-Toledo DSC1 analyser from −70 °C to 250 °C. Differential scanning calorimetry analysis was performed on two samples of hydrogel dressings with different levels of vitamin C in the layers and one sample of a dressing with no added vitamin C.

#### 4.3.4. Mechanical Strength Test—Static Compression Test

The mechanical properties of PVA-based hydrogel dressings were evaluated using a Zwick Roell RetroLine tensile tester and testXpert III 1.6 software. Cylindrical specimens were subjected to uniaxial compression with an initial force of 0.03 MPa, a compression module speed of 1 mm/min and a test speed of 5 mm/min.

#### 4.3.5. Membrane Diffusion

In order to simulate the action of a hydrogel dressing, a custom design was created that functioned as a membrane. After placing the dressings in the container cap, they were immersed in a solution containing 80 mL of distilled water so that the dressing was immersed in water from below (Figure 19). The samples prepared in this way were then placed in a laboratory dryer where a constant temperature of 35 °C was maintained. During the study, changes in pH and changes in conductivity were measured over time as a function of the vitamin C content of the dressing layers.

## Figures and Tables

**Figure 1 ijms-25-10565-f001:**
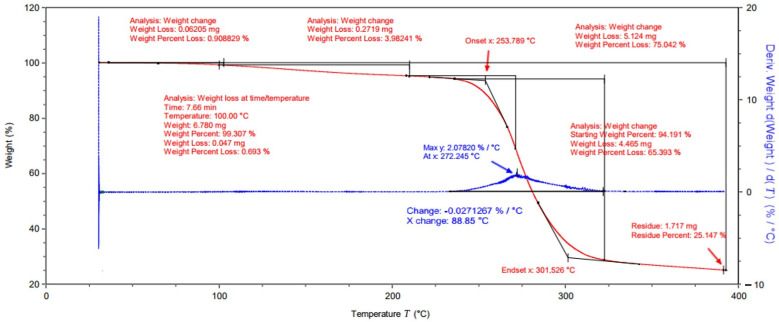
Thermogravimetric analysis for PVA.

**Figure 2 ijms-25-10565-f002:**
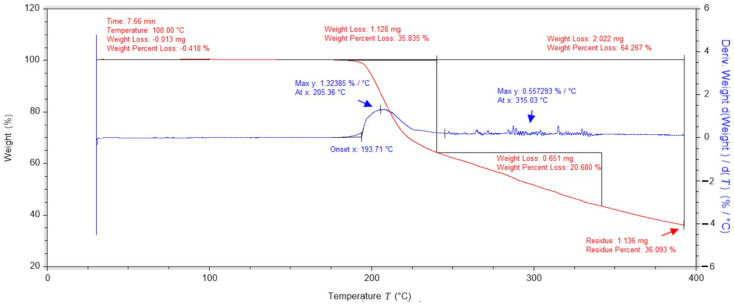
Thermogravimetric analysis of vitamin C.

**Figure 3 ijms-25-10565-f003:**
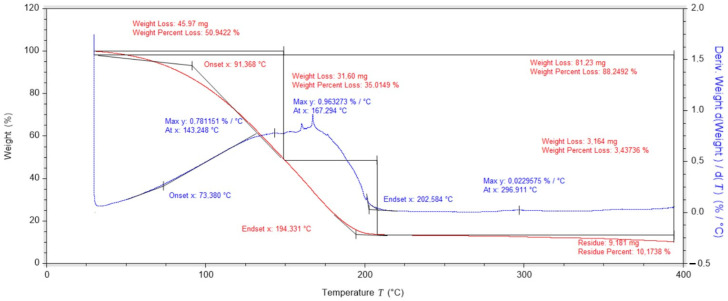
Thermogravimetric analysis of a dressing layer containing PVA, nanotubes and gauze.

**Figure 4 ijms-25-10565-f004:**
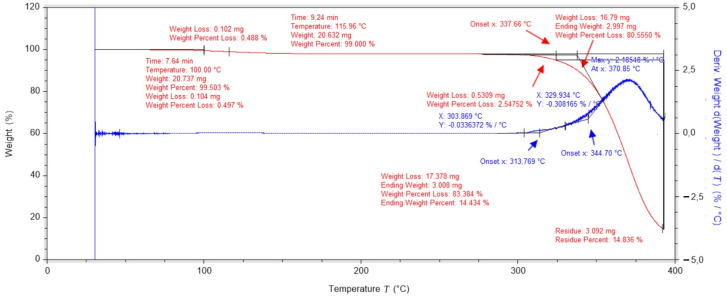
Thermogravimetric analysis of a PLA-containing dressing layer with cotton gauze.

**Figure 5 ijms-25-10565-f005:**
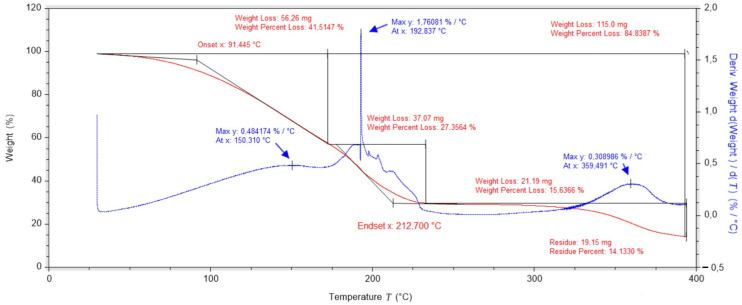
Thermogravimetric analysis of a dressing without vitamin C.

**Figure 6 ijms-25-10565-f006:**
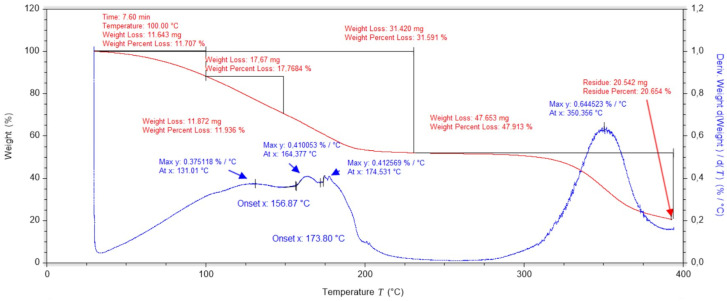
Thermogravimetric analysis of a vitamin C dressing in two hydrogel layers.

**Figure 7 ijms-25-10565-f007:**
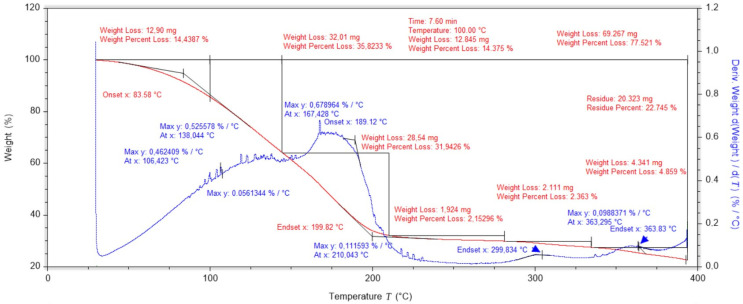
Thermogravimetric analysis of a vitamin C dressing in one hydrogel layer.

**Figure 8 ijms-25-10565-f008:**
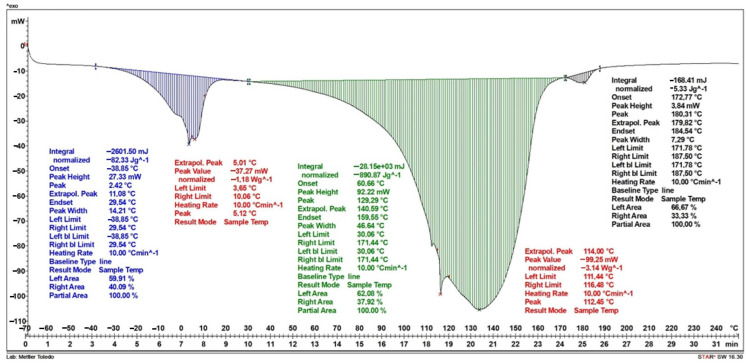
DSC curve of the dressing without added vitamin C.

**Figure 9 ijms-25-10565-f009:**
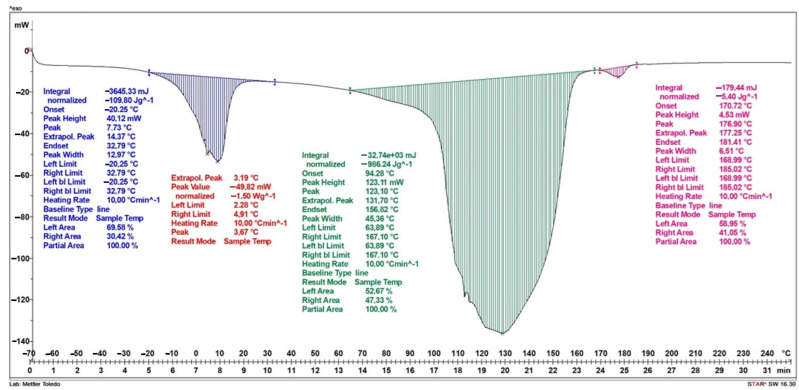
DSC curve of the dressing containing vitamin C in both hydrogel layers.

**Figure 10 ijms-25-10565-f010:**
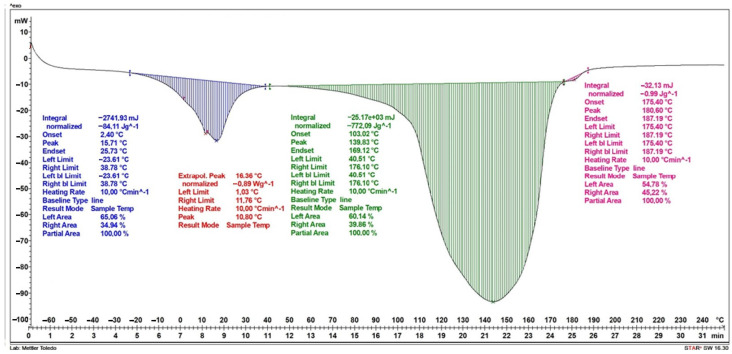
DSC curve of a dressing containing vitamin C in one hydrogel layer.

**Figure 11 ijms-25-10565-f011:**
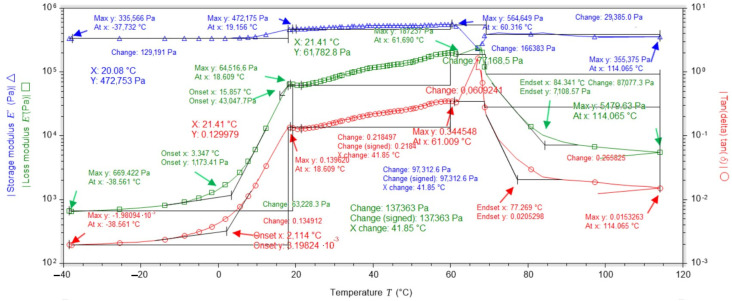
DMA test results for the dressing without added vitamin C.

**Figure 12 ijms-25-10565-f012:**
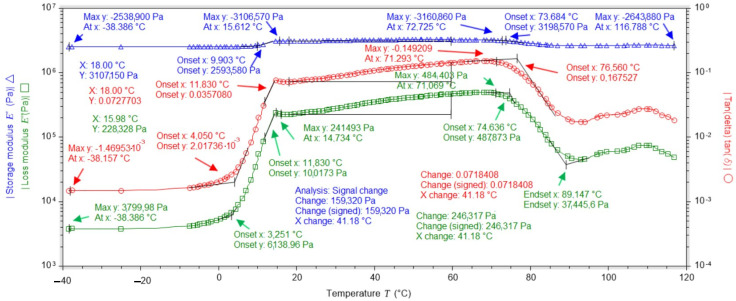
DMA test results for the dressing with added vitamin C in both hydrogel layers.

**Figure 13 ijms-25-10565-f013:**
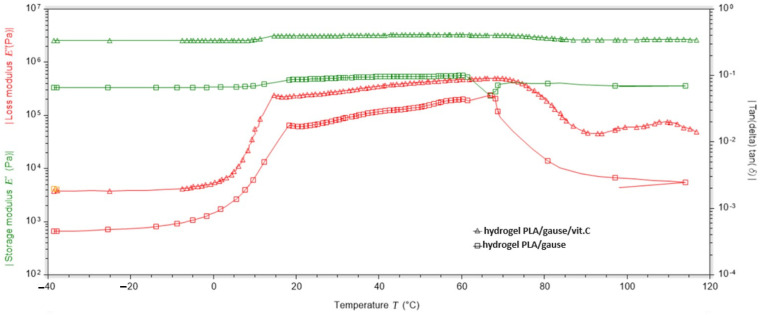
Summary of the results of the elastic modulus and loss modulus tests.

**Figure 14 ijms-25-10565-f014:**
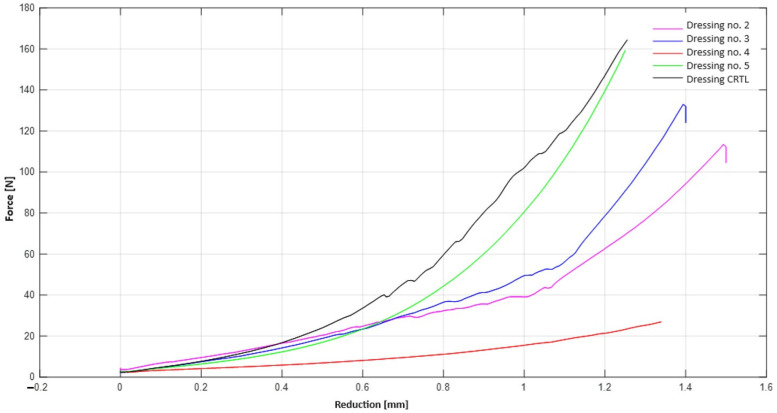
Summary of compression test results for the prepared dressings.

**Figure 15 ijms-25-10565-f015:**
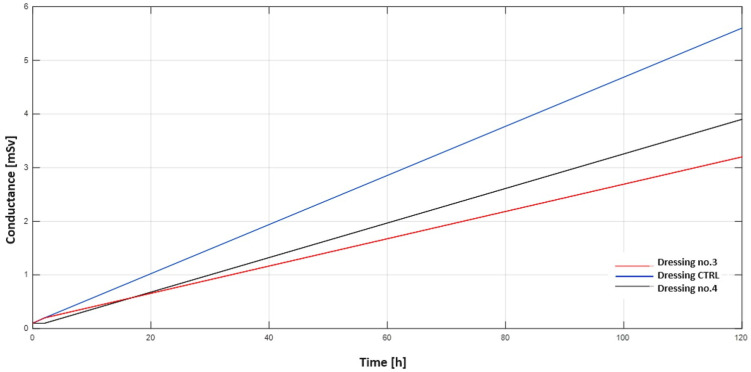
Graph of conductance change for three different hydrogel dressings.

**Figure 16 ijms-25-10565-f016:**
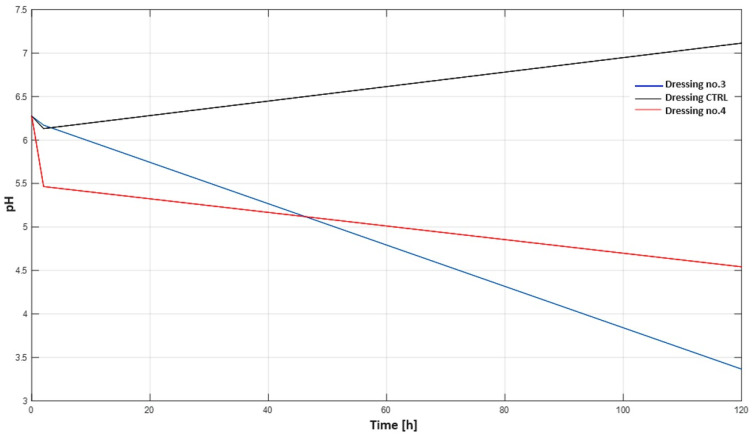
Graph of pH change for three different hydrogel dressings.

**Figure 17 ijms-25-10565-f017:**
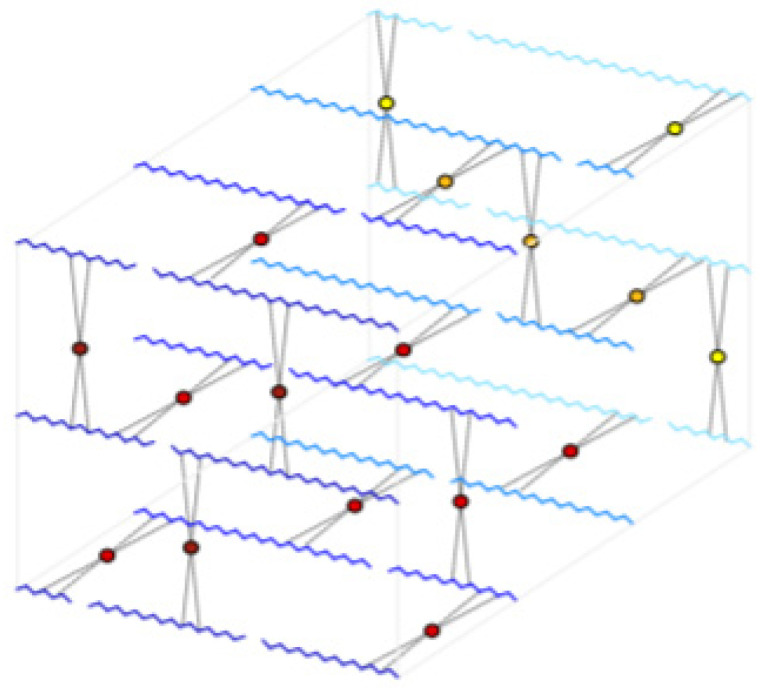
Scheme of spatial connection of PVA chains using borate ion molecules [20].

**Figure 18 ijms-25-10565-f018:**
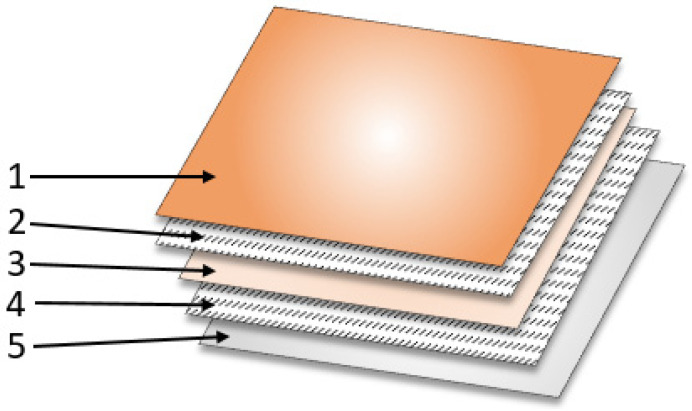
Dressing scheme.

**Figure 19 ijms-25-10565-f019:**
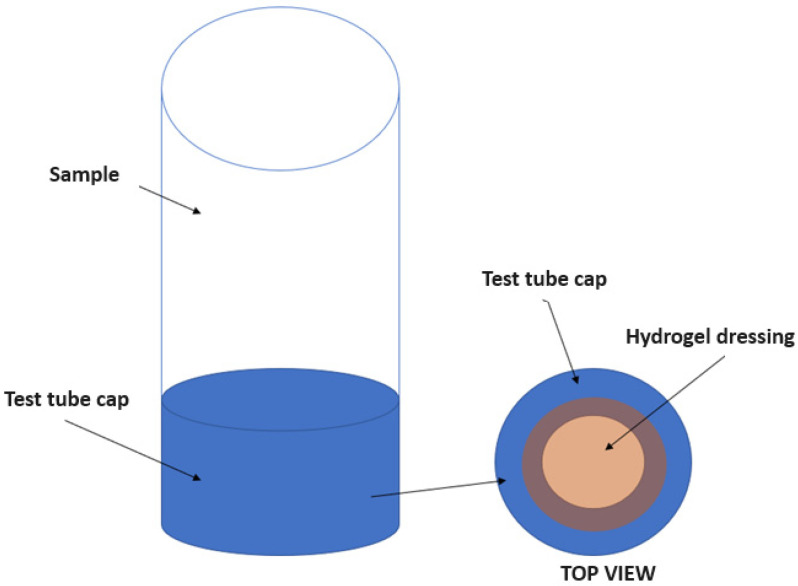
Diagram of how to place the hydrogel in a test tube.

**Table 1 ijms-25-10565-t001:** Summary of results obtained from the TG study.

No.	Sample Name	T_1%_ [°C]	T_2%_ [°C]	T_3%_ [°C]	T_4%_ [°C]	T_5%_ [°C]	Residue in 400 °C [%]
1.	PVA	110	133	156	184	220	25.14 ± 0.45
2.	PLA + gauze	116	165	311	318	323	14.83 ± 0.34
3.	PVA + gauze + nanotubes	43	51	58	63	67	10.156 ± 0.15
4.	Vitamin C	194	196	197	198	199	36.09 ± 0.62
5.	PVA + PLA ctrl	30	50	61	68	74	14.11 ± 0.37
6.	PVA + PLA 2	47	56	64	70	75	20.65 ± 0.43
7.	PVA + PLA 3	45	54	60	66	71	22.74 ± 0.49

**Table 2 ijms-25-10565-t002:** Summary of characteristic values from the DSC analysis run.

No.	Sample Name	First Degree				Second Degree				Third Degree			
		T_0_ [°C]	T_k_ [°C]	T_max_ [°C]	S [J/g]	T_0_ [°C]	T_k_ [°C]	T_max_ [°C]	S [J/g]	T_0_ [°C]	T_k_ [°C]	T_max_ [°C]	S [J/g]
1.	Dressing-ctrl	−38.9	29.5	2.4	82.3	30.1	171.4	129.3	890.9	171.9	187.5	180.3	5.33
2.	Dressing no. 2	−20.3	32.8	7.7	109.8	94.3	167.1	123.1	986.2	170.7	181.4	176.9	5.40
3.	Dressing no. 3	−23.6	38.8	15.7	84.1	40.5	176.1	139.8	772.1	175.4	187.2	180.6	0.99

**Table 3 ijms-25-10565-t003:** Comparison of the values of the modulus of elasticity E’ and the loss modulus E’’ at characteristic temperatures.

No.	Sample Name	Elastic Modulus [Pa] for:		Loss Modulus [Pa] for:	
		18 °C	60 °C	18 °C	60 °C
1.	Without vitamin C	464,199	562,676	61,291.5	199,166
2.	With vitamin C	3,107,150	3,266,330	226,109	473,447

**Table 5 ijms-25-10565-t005:** Measurements obtained from the study.

No.	Sample Name	Measurement	Time					
			0	2 h	24 h		24 h	7th Day
1.	Dressing-ctrl	pH	6.275	6.131	7.114	Change of water in the sample	7.373	7.692
		Conductance [mSv]	0.2	0.1	3.3		0.1	0.2
2.	Dressing no. 4	pH	6.275	5.465	4.542		7.361	6.856
		Conductance [mSv]	0.2	0.2	3.2		2.1	2.1
3.	Dressing no. 3	pH	6.725	6.117	3.366		6.317	7.295
		Conductance [mSv]	0.2	0.2	5.6		0.1	1.3

## Data Availability

Data is contained within the article.

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
