# Peer review of "Hydrogel Dressing Biomaterial Enriched with Vitamin C: Synthesis and Characterization"

_ijms, 2024, doi:10.3390/ijms251910565_

Round 1

Reviewer 1 Report

Comments and Suggestions for Authors

The manuscript aimed to produce biomaterials based on hydrogel enriched with ascorbic acid in order to be used as a dressing for hard-to-heal wounds. The produced materials were characterized by thermogravimetric analysis (TG), dynamic mechanical analysis (DMA), differential scanning calorimetry (DSC), mechanical strength test, membrane test, testing the stability of the samples. The manuscript is oriented towards an applicative character.

1) Although the authors use several methods of characterization, X-Ray powder diffraction should have been included because with this method the structural transformations that take place following some processes or treatments are highlighted and also the degree of crystallinity of the samples.

2) The initial compositions of the samples should have been specified better.

3) Thermogravimetric analysis and DSC should be written one after the other or together to be correlated.

4) The conclusions must be summarized.

5) The originality of the research must be highlighted

Author Response

 (Reviewer 1)

Thank you very much for your comments.

Comments and Suggestions for Authors:
The manuscript aimed to produce biomaterials based on hydrogel enriched with ascorbic acid in order to be used as a dressing for hard-to-heal wounds. The produced materials were characterized by thermogravimetric analysis (TG), dynamic mechanical analysis (DMA), differential scanning calorimetry (DSC), mechanical strength test, membrane test, testing the stability of the samples. The manuscript is oriented towards an applicative character.

  • Although the authors use several methods of characterization, X-Ray powder diffraction should have been included because with this method the structural transformations that take place following some processes or treatments are highlighted and also the degree of crystallinity of the samples.

Thank you for your suggestion regarding possible studies of our hydrogel dressings. XRD studies are very interesting, while the additives tested in our proposed dressing, although in their original form they are powders, are ultimately soluble in water and in the structure of the dressing are dispersed in the aqueous matrix of the dressing. The authors do not expect a crystalline phase present in the created dressings due to the fact that the structure of these dressings is on the border between liquid and solid. Water effectively pushes the PVA chains (the scaffolds of the hydrogel dressing) apart, and therefore we do not expect an ordered crystalline structure to appear there. In our studies, we focused on the aspect of thermodynamic hydrogel structures.

  • The initial compositions of the samples should have been specified better.

Thank you for this comment, we have corrected the description of the layer configurations of the tested dressings - it was indeed a bit illegible. As suggested, we have described exactly what layers are included in each configuration –
“From the prepared layers, four types of dressings were made with different vita-min C content in the layers:

  • Dressing No. 1 – no vitamin C content in the first and third layers (control dressing)

/1st layer - pure hydrogel/2nd layer - hydrogel with CNT and gauze/3rd layer - pure hydrogel/4th layer - hydrogel with CNT and gauze/5th layer - hydrogel with CNT and gauze/PLA film with gauze

  • Dressing No. 2 – vitamin C content in the first and third layers

/1st layer - hydrogel with vitamin C/2nd layer - hydrogel with CNT and gauze/3rd layer - hydrogel with vitamin C/4th layer - hydrogel with CNT and gauze/5th layer - hydrogel with CNT and gauze/PLA film with gauze/

  • Dressing No. 3 – vitamin C content in the first layer

/1st layer - hydrogel with vitamin C/2nd layer - hydrogel with CNT and gauze/3rd layer - pure hydrogel/4th layer - hydrogel with CNT and gauze/5th layer - hydrogel with CNT and gauze/PLA film with gauze/

  • Dressing No. 4 – vitamin C content in the third layer

/1st layer - pure hydrogel/2nd layer - hydrogel with CNT and gauze/3rd layer - hydro-gel with vitamin C/4th layer - hydrogel with CNT and gauze/5th layer - hydrogel with CNT and gauze/PLA film with gauze/”

  • Thermogravimetric analysis and DSC should be written one after the other or together to be correlated.

Thank you for this valid comment, we have changed the order of presentation of the results. After TG studies we present DSC studies. It is more legible now.

4) The conclusions must be summarized.

5) The originality of the research must be highlighted

Answer to point 4) and 5)

Thank you for your valuable comments. The most important conclusions and perspectives were summarized, and the innovative nature of the research was also emphasized - “In summary, the studies have shown that the properties of dressings differ de-pending on the number of layers with vitamin C and also on the location of the layers with vitamin C in the dressing. In general, the proposed layered dressings are charac-terized by high stability, durability and effective release of ions into the wound envi-ronment. The presented studies are oriented towards an applicative character. How-ever, further research should be carried out to produce the most optimal, in terms of usability and effectiveness, design of a layered hydrogel dressing with ascorbic acid. Currently, hydrogel dressings for wounds are available on the market, but the authors have no knowledge about the available hydrogel dressings with ascorbic acid used for difficult-to-heal wounds. The designed layered hydrogel dressings enriched with vita-min C seem to be an innovative solution and may be groundbreaking on the market, due to low costs and ease of production and due to the effectiveness of supporting the treatment of difficult-to-heal wounds.”

The manuscript has been checked for correctness of the English language.

Thank you very much for your time.

Reviewer 2 Report

Comments and Suggestions for Authors

The manuscript titled "Hydrogel dressing biomaterial enriched with a vitamin C: synthesis and characterization" deals with the preparation and through characterization of a set of hydrogels containing ascorbic acid.

The manuscript is very detailed and provides lots of data on the thermal behavior and mechanical properties of the prepared systems.

The introduction provides the information that is needed in order to put the manuscript in the right perspective. I strongly recommend adding references to the few first statements of the introduction. I also suggest avoiding the term ""proved" as it has no place in a scientific paper. One can, at most, show, support a hypothesis, disprove, and alike.

Another general point I have is with significant figures and error bars. If an experiment is done more than once, I expect the authors to disclose how many times it was done and what is the value, and what is the error bar. If not, please disclose values using rounded numbers. I have some experience with TGA, DSC and DMA, and weight loss of 74.853% usually means 75+/-X% at best. Same is with many of the values stated in the manuscript.

The manuscript contains some places where things are written in Polish (probably, surely not English). This needs to be corrected.

Yet another general Issue is with the overall structure of the manuscript. It contains lots of data, limited interpretation and no vision. I strongly recommend the authors to deal with this point as without it, this paper will be carried away with the huge paper stream and leave no mark.

Comments on the Quality of English Language

The manuscript deserves some final polishing.

Author Response

 (Reviewer 2)

Thank you very much for your comments.

Comments and Suggestions for Authors:
The manuscript titled "Hydrogel dressing biomaterial enriched with a vitamin C: synthesis and characterization" deals with the preparation and through characterization of a set of hydrogels containing ascorbic acid.

The manuscript is very detailed and provides lots of data on the thermal behavior and mechanical properties of the prepared systems.

  • The introduction provides the information that is needed in order to put the manuscript in the right perspective. I strongly recommend adding references to the few first statements of the introduction. I also suggest avoiding the term "proved" as it has no place in a scientific paper. One can, at most, show, support a hypothesis, disprove, and alike.

Thank you for your valuable comments. As suggested, the introduction has been revised and missing references have been added. Throughout the manuscript, the words "proven" have been replaced with more appropriate ones – “Difficult-to-heal and chronic wounds are a problem that has always accompanied people [1]. Wound treatment is associated with chronic pain and discomfort for patients [1]. Currently, due to the prevalence of civilization diseases and an aging population, the problem of chronic wounds such as diabetic wounds or pressure ulcers is growing. To accelerate the wound healing process, initially plant, herbal preparations or natural substances containing primarily animal fat, milk, honey or egg whites were used [1, 2]. In the 20th century, George D. Winter made a breakthrough in the history of dressings. He showed that the use of an active dressing in a moist environment doubles the speed of wound healing [3]. This discovery has led to the development of many generations of wound dressings, and the latest third-generation dressings are based on hydrogels and hydrophilic polyurethane films [3]. Currently, poor wound healing can be caused by many factors, such as post-operative complications. Impaired wound healing is common in patients with diabetes, in the elderly with leg ulcers and in patients with pressure ulcers [3]. Furthermore, non-healing or slow-healing wounds have a significant impact on the patient's quality of life, causing pain and discomfort, and often leading to social isolation, depression and financial stress.”

  • Another general point I have is with significant figures and error bars. If an experiment is done more than once, I expect the authors to disclose how many times it was done and what is the value, and what is the error bar. If not, please disclose values using rounded numbers. I have some experience with TGA, DSC and DMA, and weight loss of 74.853% usually means 75+/-X% at best. Same is with many of the values stated in the manuscript.

Thank you for pointing out this important issue. We have corrected the statistics in the manuscript.

  • The manuscript contains some places where things are written in Polish (probably, surely not English). This needs to be corrected.

Thank you very much for this valuable comment. We have corrected Polish words to English words - Table 3. and Table 5.

  • Yet another general Issue is with the overall structure of the manuscript. It contains lots of data, limited interpretation and no vision. I strongly recommend the authors to deal with this point as without it, this paper will be carried away with the huge paper stream and leave no mark.

Thank you very much for your suggestion. A paragraph has been added to the manuscript, which summarise the most important conclusions and future perspectives, and also emphasizing the novelty of the research - “In summary, the studies have shown that the properties of dressings differ de-pending on the number of layers with vitamin C and also on the location of the layers with vitamin C in the dressing. In general, the proposed layered dressings are characterized by high stability, durability and effective release of ions into the wound environment. The presented studies are oriented towards an applicative character. How-ever, further research should be carried out to produce the most optimal, in terms of usability and effectiveness, design of a layered hydrogel dressing with ascorbic acid. Currently, hydrogel dressings for wounds are available on the market, but the authors have no knowledge about the available hydrogel dressings with ascorbic acid used for difficult-to-heal wounds. The designed layered hydrogel dressings enriched with vita-min C seem to be an innovative solution and may be groundbreaking on the market, due to low costs and ease of production and due to the effectiveness of supporting the treatment of difficult-to-heal wounds.”

The manuscript has been checked for correctness of the English language.

 Thank you very much for your time.

Round 2

Reviewer 1 Report

Comments and Suggestions for Authors

The manuscript can be accepted in its current form